# Graphene-Based Thermal Interface Materials: An Application-Oriented Perspective on Architecture Design

**DOI:** 10.3390/polym10111201

**Published:** 2018-10-27

**Authors:** Le Lv, Wen Dai, Aijun Li, Cheng-Te Lin

**Affiliations:** 1School of Materials Science and Engineering, Shanghai University, Shanghai 200444, China; lvle@nimte.ac.cn; 2Key Laboratory of Marine Materials and Related Technologies, Zhejiang Key Laboratory of Marine Materials and Protective Technologies, Ningbo Institute of Materials Technology and Engineering, Chinese Academy of Sciences, Ningbo 315201, China; daiwen@nimte.ac.cn

**Keywords:** graphene, thermal interface materials, application-orientation

## Abstract

With the increasing power density of electrical and electronic devices, there has been an urgent demand for the development of thermal interface materials (TIMs) with high through-plane thermal conductivity for handling the issue of thermal management. Graphene exhibited significant potential for the development of TIMs, due to its ultra-high intrinsic thermal conductivity. In this perspective, we introduce three state-of-the-art graphene-based TIMs, including dispersed graphene/polymers, graphene framework/polymers and inorganic graphene-based monoliths. The advantages and limitations of them were discussed from an application point of view. In addition, possible strategies and future research directions in the development of high-performance graphene-based TIMs are also discussed.

## 1. Introduction

Thermal management has become a central issue to guarantee the reliability and service life of electrical and electronic devices, such as highly integrated CPUs, high-power LED lights and energy harvesting systems [1]. Particularly, thermal interface materials (TIMs) are an important and indispensable part for the efficient transfer/removal of generated heat from heaters (e.g., semiconductor chips) to avoid electronic devices working under overheating conditions [2]. In actual working status (Figure 1), TIMs with high through-plane thermal conductivity and good compressibility bridge the heater and heat sink, filling the unavoidable air gaps between their mating interface [3,4]. Conventional TIMs are made up of polymer matrices filled by thermally conductive materials (boron nitride, aluminum nitride, alumina, etc.) to achieve the thermal conductivity of 1–5 W/mK (50–70 wt % filler loading) [5,6,7]. However, with the shrinking feature size and improving power density of electronic devices, the accompanying thermal management has gone beyond the processing capacity of conventional TIMs [3]. Therefore, the development of new generational TIMs with improved thermal conductivity has drawn extensive attention and devotion for many researchers.

Since graphene was discovered and exhibited an ultra-high intrinsic thermal conductivity of 3500–5300 W/mK [8,9], a large amount of academic interest has been focused on the development of high thermally conductive graphene-based TIMs. In this perspective, we highlight three types of graphene-based TIMs, including dispersed graphene/polymers, graphene framework/polymers and inorganic graphene-based monoliths. The advantages and limitations of them were discussed from an application point of view. By means of the points raised in this perspective, we hope to provide some assistance for the further development of TIMs.

## 2. Graphene-Based TIMs

### 2.1. Dispersed Graphene/Polymers

In earlier research, graphene-based TIMs were prepared through a solution or melt-blending process to disperse graphene sheets into a polymer matrix (Figure 2a). A summary of thermal conductivity (κ) of the dispersed graphene/polymers is listed in Table 1. As seen from the table, to get the desired thermal conductivity beyond conventional TIMs (5 W/mK), 20–50 wt % of graphene loading in polymer is essential. However, such high graphene content usually results in high viscosity for printing operations and poor mechanical properties for the cured composites, hardly meeting the demands of current industry for practical applications [10]. One other thing to note is that there has been no obvious improvement in thermal conductivity of dispersed graphene/polymers during the last decade. This less than satisfactory progress is mainly attributed to two factors, firstly the difficulty of graphene dispersion in a polymer matrix (Figure 2b) as a result of the strong π–π interaction between graphene sheets, and secondly, the strong interfacial phonon scattering of the graphene and polymer leading to the low effective thermal conductivity of the graphene filler [11,12,13]. Even so, the development of oriented alignment and surface modification technology of graphene filler may break this bottleneck.

### 2.2. Graphene Framework/Polymers

In recent years, the penetration of polymers into three-dimensional graphene frameworks has been considered as a promising solution to develop composites with improved thermal conductivity for TIM applications. So far, the approaches for the fabrication of a graphene framework can be mainly classified into two categories, the self-assembly method and the template-synthesis method [21,22,23,24]. The former is usually carried out by the self-assembly and hydrothermal reduction of graphene oxide (GO) in aqueous solution followed by freeze-drying or air-drying to achieve the graphene framework, as shown in Figure 3a. The latter is performed by coating graphene sheets onto the skeleton of porous sponge (e.g., polyurethane foam) or growing the graphene on the nickel (Ni) foam through chemical vapor deposition (CVD), followed by the removal of the sponge and Ni to obtain the graphene framework (Figure 3b). Owing to the formation of continuous thermal transport paths within the polymer matrices, graphene frameworks/polymers exhibit relatively higher thermal conductivity in comparison to their dispersed graphene/polymer counterparts at a similar filler content [25]. Existing research on graphene framework/polymers can be divided into two types according to the hardness of the matrix: Rigid plastics (e.g., epoxy as matrix) and soft silicon rubber (e.g., polydimethylsiloxane (PDMS) as a matrix). Recent literature based on graphene framework/epoxy composites is summarized in Table 2, suggesting great progress in the improvement of thermal conductivity of composites. Particularly, the highest thermal conductivity of up to 35.5 W/mK was reported for the epoxy containing a densely packed and vertically aligned graphene framework, which was synthesized by hydrothermally treating the suspension of GO and graphene followed by air-drying and annealing, as shown in Figure 3a [26]. For the performance indicators of TIMs, however, high thermal conductivity is just one of the necessary factors. As is known to all, TIMs are used to fill the gaps between the heater and heat sink (Figure 1b), so deformability for TIMs under the packaging conduction is also an essential feature [2]. However, different from the dispersed graphene/epoxy composite that can be packaged by solution precursor coating followed by curing, the reported graphene framework/epoxy composites are all solidified hard bulk and the corresponding compressive modulus is up to 2–5 GPa [27,28], similar to metal and graphite, inherently limiting their application as TIMs.

In contrast to hard epoxy, soft PDMS with the lower compressive modulus of 2–10 MPa [32,33] is a suitable material to address the challenge for gap filling. However, the viscosity of PDMS precursor (3900–4150 mPa·s [34]) is much higher than that of epoxy (200–400 mPa·s [35]), resulting in difficulty in PDMS penetration into the graphene framework. In general, the penetration depth of PDMS is proportional to the pore size of the graphene framework. Therefore, two kinds of graphene framework were commonly employed to incorporate with PDMS for the fabrication of homogeneous composites: Macroporous graphene framework and densely packed graphene framework with lower thickness. The former usually has a low density (5–10 mg/cm^3^) [36], such as the Ni foam templated graphene framework shown in Figure 3c, leading to a very low addition amount of graphene into PDMS (less than 2 wt %). Correspondingly, the thermal conductivity of the resulting composites is less than 1.5 W/mK (see Table 3). For the case of a densely packed graphene framework, a widely reported preparation process is to compress the Ni foam templated graphene framework in the vertical direction, by which a porous graphene film (thickness of 180–1000 μm and density of 95–140 mg/cm^3^) composed of densely packed CVD graphene can be obtained, as shown in Figure 3d [25,37]. In Bai’s work, the corresponding graphene framework/PDMS composite (Figure 3e) exhibited a high thermal conductivity of 28.77 W/mK (11.6 wt %) along the in-plane direction [37]. However, owing to the rearrangement of graphene towards the in-plane direction, the resulting through-plane thermal conductivity is as low as 1.62 W/mK, which can be further improved (2.11 W/mK) by modifying boron nitride (BN) on the graphene surface [38]. The value is lower than that of commercial TIMs (5 W/mK), resulting in insufficiency for the heat transfer demand along the through-plane direction of TIMs in practical application.

Most recently, we reported a filtrated graphene framework composed of a horizontal and densely packed graphene structure with a density of 60–80 mg/cm^3^ [31]. In order to meet the through-plane heat dissipation demands of TIM applications and solve the penetration issue of PDMS, as shown in Figure 4a, we cut the as-prepared graphene framework into thin slices along the vertical direction and flipped the slices by 90 degrees to obtain a vertically aligned and densely packed graphene framework with a thickness of 1 mm (Figure 4b). After incorporating with PDMS, the composite presents the through-plane thermal conductivity of 11.7 W/mK (7 wt %), showing a promising application as a TIM. In the actual packaging process, to guarantee a good gap filling, a compression along the vertical direction on TIMs is inevitable, resulting in the deformation of samples (Figure 1b). Accordingly, the thermal conductivity of our graphene framework/PDMS composite after compressing (strain: 50%) was studied with the resulting value decreasing by 60% (3.5 W/mK). This decrease can be attributed to the fact that the vertically aligned and densely packed graphene framework is an inferior structure to realize reversible deformation under compression, which results in the fractures of the framework [21], as shown in Figure 4a,c. The comparative microscopic morphology of composites before (Figure 4d) and after (Figure 4e) compressing shows that the graphene sheets were detached from the matrix and many cracks were formed within the composite that had undergone the compression, leading to the decrease of thermal conductivity.

As discussed above, graphene frameworks show great potential for the devotion of polymer composites with improved thermal conductivity. So far, however, there are few reports on graphene framework/polymer composites for TIM application with the thermal management performance in excess of commercial TIMs. Part of the reason is that the removal of heat through the interface requires TIMs to have high through-plane thermal conductivity, while the high thermal conductivity of graphene is only provided along the in-plane direction [4,8]. In order to achieve a practical TIM by combining graphene framework and polymer for the next generation of thermal management application, four fundamental requirements need to be met:(i)Soft elastomers as matrix for good gap filling, such as PDMS.(ii)A relatively higher density of the graphene framework for higher graphene content in the polymer matrix, leading to high thermal conductivity.(iii)An opportune arrangement of graphene in the framework for avoiding an unsatisfactory thermal conductivity in the through-plane direction.(iv)Graphene framework with reversible deformation for maintaining high thermal conductivity while under packaging.

### 2.3. Inorganic Graphene-Based Monoliths

Apart from the incorporation with polymers, graphene can also be used as a TIM by the fabrication of inorganic graphene-based monoliths. In 2011, Wong and co-workers reported a graphene monolith (1.6 g/cm^3^) composed of vertically aligned graphene sheets, which was constructed via “rotating-reassembling” as-prepared graphene paper (Figure 5a–c). With an indium coating on the surface, the obtained sample presents an equivalent through-plane thermal conductivity of 75.5 W/mK and contact thermal resistant of 5.1 Kmm^2^/W [44]. However, the high density close to graphite leads to its inferior deformability, constraining its wide applications. In 2016, Lv et al. fabricated a super-elastic graphene/CNT aerogel using the hydrothermal method and a subsequent freeze-drying process (Figure 5d–f). At a moderate pressure of 153.2 kPa, the hybrid graphene/CNT aerogel shows a through-plane thermal conductivity of 88.5 W/mK and contact thermal resistance of 5.1 Kmm^2^/W [45]. That same year, Teo and co-workers synthesized a Ni foam templated graphene framework (99.6% porosity, density of 1−5 mg/cm^3^) and studied the thermal conductivity as function of compression strain (Figure 5g–i). Under the vertical pressure, the sample exhibited a good mechanical property for gap filling and a high heat transfer property with a thermal conductivity of 86 W/mK. Direct comparison with commercial TIMs has shown an improved cooling performance by 20–30% [46]. According to the above description, inorganic graphene-based monoliths exhibit higher thermal conductivity than that of graphene/polymer composites, due to the elimination of high Kapitza resistance between the graphene and the polymer matrix (the interfacial thermal resistance of atomically perfect interfaces between them due to their difference in phonon frequencies) [47]. Moreover, in actual performance tests, they also show the great potential to be used as high performance TIMs. For practical application, one point to consider is that the reported inorganic graphene-based monoliths are composed of all-carbon materials, leading to the samples ease of crushing and dropping out after compression, owing to the poor binding force of the crystalline sp^2^-bonded carbon structure [48,49,50]. The slight dusting of carbon powder may cause a short circuit or contaminate some precision microelectronic and optoelectronic devices [51,52]. Perhaps the development of a nanostructured coating on the surface of inorganic graphene-based monoliths can alleviate the carbon powder pollution to solve this issue.

## 3. Conclusions and Perspectives

This report briefly reviews the progress of graphene-based TIMs and discusses three types of materials, including dispersed graphene/polymers, graphene framework/polymers and inorganic graphene-based monoliths. Each material both has its advantages and disadvantages, as shown in Table 4. Dispersed graphene/polymers with the least thermal conductivity are the most cost effective among the materials and have a similar preparation technology to conventional TIMs, showing minor obstruction towards industrial production. Inorganic graphene-based monoliths have the highest performance, but the carbon powder pollution derived from its all-carbon component limits their application in precision instruments. In comparison, by the incorporation of polymer, graphene framework/polymers have better adaptability, but the moderate thermal conductivity creates difficulty in dealing with the future development of microelectronics. In the view of application, the future direction of graphene-based materials should be focused on eliminating the disadvantages while retaining the advantages of current materials. For the researchers, besides the thermal conductivity, the contact thermal resistance and mechanical behavior of those TIM candidates need also to be considered. Actual capacities of thermal management for the next-generation TIMs are paramount.

## Figures and Tables

**Figure 1 polymers-10-01201-f001:**
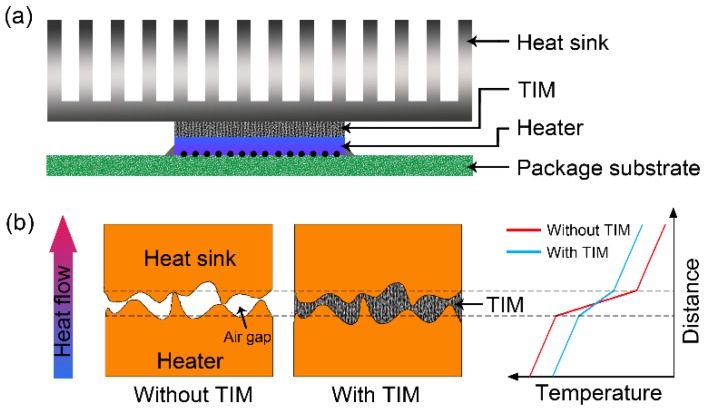
(**a**) Schematic illustrating a typical ball grid array electronics package with TIMs bridging the heater and the heat sink. (**b**) Working principle of TIMs showing that the air gaps between the mating interface of the heater and the heat sink will cause overheating of the heater and a TIM filling out the gaps can effectively reduce heater temperature.

**Figure 2 polymers-10-01201-f002:**
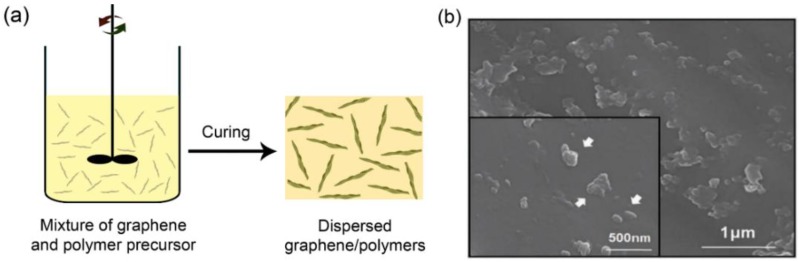
(**a**) Schematic illustrating the fabrication of dispersed graphene/polymers through a solution blending process. (**b**) Typical cross-sectional SEM image of the dispersed graphene/polymers (graphene loading: 10 wt %), showing the poor dispersion of graphene in the polymer matrix. Reprinted with permission from: (**b**) Reference [18], Copyright 2013, Wiley-VCH.

**Figure 3 polymers-10-01201-f003:**
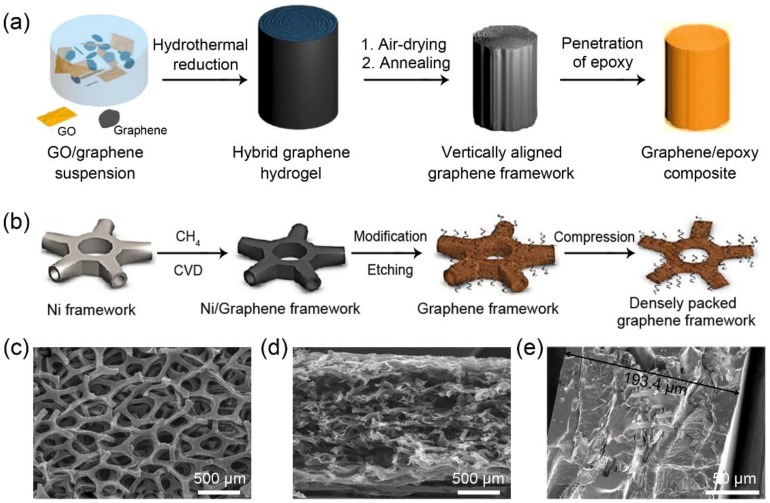
(**a**) Schematic illustrating the self-assembly method to fabricate the densely packed and vertically aligned graphene framework/polymers. (**b**) Graphene framework grown on Ni foam by chemical vapor deposition (CVD) followed by removal of Ni in the etchant. The framework was further applied a vertical compression to obtain porous graphene film composed of horizontal and densely packed CVD graphene. The typical cross-sectional SEM images of Ni foam templated graphene framework (**c**) before and (**d**) after applying a vertical compression and (**e**) Ni foam templated graphene framework/PDMS composite. Reprinted with permission from: (**a**) Reference [26], Copyright 2018, American Chemical Society; (**b**) Reference [37], Copyright 2017, American Chemical Society; (**c**,**d**) Reference [25], Copyright 2018, Royal Society of Chemistry; (**e**) Reference [38], Copyright 2017, Elsevier.

**Figure 4 polymers-10-01201-f004:**
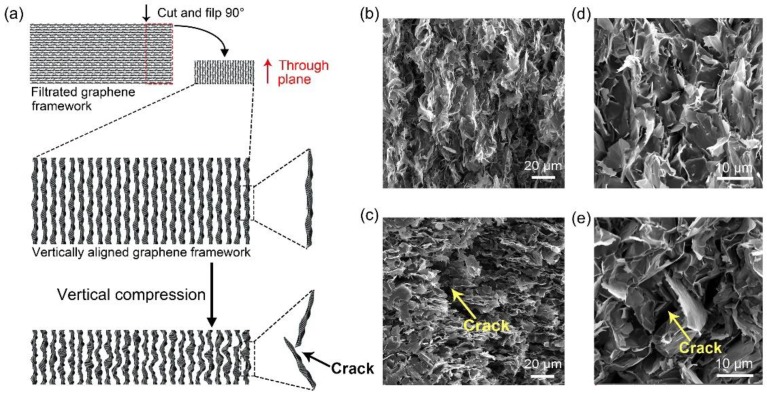
(**a**) Schematic illustrating the fabrication of vertically aligned and densely packed graphene framework and the structure of the framework was destroyed after applying a vertical compression. SEM images of vertically aligned and densely packed graphene framework (**b**) before and (**c**) after compressing. (**d**,**e**) The case of graphene framework/PDMS composite.

**Figure 5 polymers-10-01201-f005:**
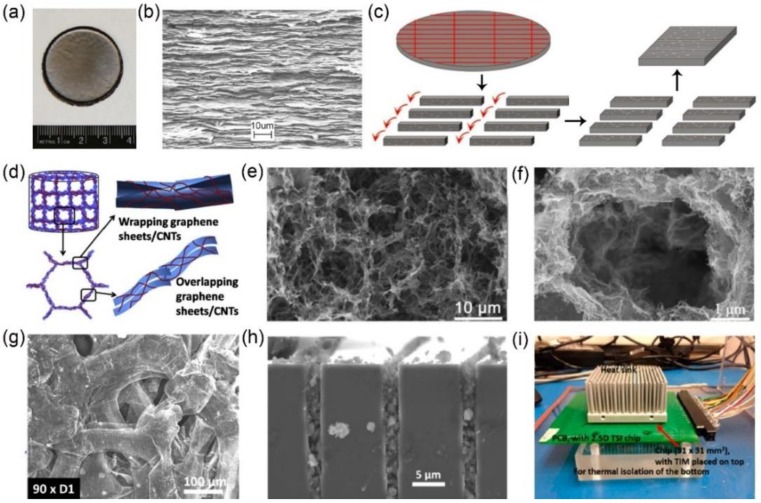
(**a**) Photograph (**b**) and SEM image of graphene paper. (**c**) Schematic illustrating the assembly of vertically aligned graphene monolith via “rotating-reassembling” as-prepared graphene paper. (**d**) Schematic illustration for the micro-structure of graphene/CNT aerogels with the SEM image showing in (**e**,**f**). (**g**) SEM image of compressed Ni foam templated graphene framework with the cross-sectional gap filling and TIM performance evaluation system showing in (**h**,**i**), respectively. Reprinted with permission from: (**a**–**c**) Reference [44], Copyright 2008, American Chemical Society; (**d**–**f**) Reference [45], Copyright 2016, Elsevier; (**g**–**i**) Reference [46], Copyright 2016, American Chemical Society.

**Table 1 polymers-10-01201-t001:** Summary of graphene/polymers prepared by a solution or melt-blending process.

Filler	Matrix	κ (W/mK)	Fraction	Years
Graphite nanoplatelet	Epoxy	6.44	≈34 wt %	2007 [14]
Functionalized exfoliated graphite	Epoxy	5.80	20 wt %	2008 [15]
Graphene + CNT	Epoxy	0.32	1 wt %	2011 [16]
Graphene + Multilayer graphene	Epoxy	5.10	≈15 wt %	2012 [7]
Graphene + CNT	Epoxy	7.30	50 wt %	2012 [17]
Functionalized graphene flakes	Epoxy	1.53	10 wt %	2013 [18]
Graphene nanoflake	Polytetrafluoroethylene	10.00	≈24 wt %	2015 [19]
Graphene nanoplatelets	Polycarbonate	7.30	20 wt %	2016 [20]
Multilayer graphene	Epoxy	1.50	5.7 wt %	2016 [11]

**Table 2 polymers-10-01201-t002:** Summary of graphene framework/epoxy composites.

Filler	κ (W/mK)	Fraction	Form	Years
Graphene aerogel	2.13 (⊥)	≈1.4 wt %	Hard bulk	2014 [10]
	0.63 (//)			
Filtrated graphene framework	16.7 (//)	≈11.8 wt%	Hard bulk	2014 [29]
	5.44 (⊥)			
Templated graphene framework	1.51	5 wt%	Hard bulk	2016 [30]
Templated graphene framework	8.80 (//)	8.3 wt%	Hard lamella	2018 [25]
	2.00 (⊥)			
Filtrated graphene framework	10.0 (//)	5.5 wt%	Hard bulk	2018 [31]
	5.40 (⊥)			
Vertically aligned graphene framework	17.2 (//)	≈33 wt%	Hard bulk	2018 [26]
	35.5 (⊥)			

⊥: through-plane thermal conductivity, //: in-plane thermal conductivity.

**Table 3 polymers-10-01201-t003:** Summary of graphene framework/polydimethylsiloxane (PDMS) composites.

Filler	κ (W/mK)	Fraction	Form	Years
Graphene aerogel	0.68	1 wt %	Macroporous	2015 [39]
Graphene aerogel	0.56	0.7 wt %	Macroporous	2015 [40]
Graphene aerogel + graphene flakes	1.08	≈0.4 + 5.3 wt %	Macroporous	2016 [41]
Graphene aerogel + carbon fiber	0.55	≈1.1 + 19.4 wt %	Macroporous	2016 [42]
Graphene aerogel	0.82	2 wt %	Macroporous	2017 [43]
Templated graphene framework	1.62 (⊥)	11.6 wt %	Densely packed	2017 [37]
	28.8 (//)			
Templated graphene framework + BN	2.11 (⊥)	33.8 wt %	Densely packed	2017 [38]
	23.5 (//)			

⊥: through-plane thermal conductivity, //: in-plane thermal conductivity.

**Table 4 polymers-10-01201-t004:** Overview of three key parameters (thermal conductivity, cost and adaptability) of dispersed graphene/polymers, graphene framework/polymers and inorganic graphene-based monoliths for analyzing their advantages and disadvantages form application point of view.

Graphene-Based TIMs	Thermal Conductivity	Cost	Adaptability
Dispersed graphene/polymers	Low	Low	Good
Graphene framework/polymers	Medium	High	Good
Graphene-based monoliths	High	High	Medium

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
