# Peer review of "Graphene-Based Thermal Interface Materials: An Application-Oriented Perspective on Architecture Design"

_polymers, 2018, doi:10.3390/polym10111201_

Round 1
Reviewer 1 Report
The manuscript "Graphene-Based Thermal Interface Materials: An Application-Oriented Perspective on Architecture Design" gives a clear perspective for those graphene polymer materials. The manuscript is very well written.
There only minor points to address
Page 1, introduction, line 26: what means (e.g. Si dies)?
Page 1, introduction, line 30, (BN, AlN, Al2O3, etc.) please define those abbreviations
Page 5, Table 2 what the meaning of the signs behind the κ (W/mK), please define
Page 7, Line 199: What means Kapitza resistence. it would be helpful for readers to give a short explanation
Author Response
Dear editor,
We are very grateful to your and the reviewers’ critical comments and thoughtful suggestions. Based on these comments and suggestions, we have made careful revision on the original manuscript entitled "Graphene-Based Thermal Interface Materials: An Application-Oriented Perspective on Architecture Design" (Manuscript ID: polymers-379385). We responded point by point to the comments of reviewer #1 as listed below, along with a clear indication of the location of the revision.
In specific response to the points raised by reviewer #1:
Question 1: Page 1, introduction, line 26: what means (e.g. Si dies) ?.
Reply: Silicon die (Si die) means semiconductor chips machined using silicon as raw material. Based on your comment, we have changed the “Si die” into “semiconductor chips” for making the manuscript easier for the reader to follow. (See Line 29, Page 1)
Question 2: Page 1, introduction, line 30, (BN, AlN, Al2O3, etc.) please define those abbreviations.
Reply: Thank you for your suggestion. The abbreviations (BN, AlN, Al2O3, etc.) have be transformed into the full name (boron nitride, aluminum nitride, alumina, etc.). (See Line 33, Page 1)
Question 3: Page 5, Table 2 what the meaning of the signs behind the κ (W/mK), please define.
Reply: The signs behind the thermal conductivity (κ) define the direction of the thermal conductivity. The sign “⊥” means the through-plane thermal conductivity and “∥” means the in-plane thermal conductivity of the sample. These signs were widely used in academic literature based on thermal management (such as J. Mater. Chem. A, 2018, 6,12091). Based on your suggestion, the explanation of these signs was added behind the Table 2 and 3. (See Line 143 and 146, Page 5)
Question 4: Page 7, Line 199: What means Kapitza resistance. it would be helpful for readers to give a short explanation.
Reply: Thank you for your suggestion. Kapitza resistance is the interfacial thermal resistance of atomically perfect interfaces between different materials. When an energy carrier (phonon or electron, depending on the material) attempts to traverse the interface, it will scatter at the interface and lead to Kapitza resistance, due to differences in electronic and vibrational properties in different materials. A discussion has been added in the manuscript (Line 206 – 207, Page 7).
We appreciate for your warm work earnestly, and hope that the correction will meet with approval. The manuscript has been overall checked, and the changes marked in red. We hope that these revisions are sufficient to make our manuscript acceptable for publication in Polymers. If you believe that any additional clarifications need to be addressed, I will be happy to include them. Once again, thank you very much for your comments and suggestions.
Yours sincerely,
Cheng-Te Lin
Ningbo Institute of Materials Technology and Engineering, Chinese Academy of Sciences
E-mail: linzhengde@nimte.ac.cn

Reviewer 2 Report
The paper from Lv et al. reports on the state of the art of graphene-based interface materials, highlighting their advantages, as well as their current limitations in view of their possible exploitation. The paper is well written and gives a good overall "picture" of the state of the art related to the topic. Minor concerns refer to:
- Tables 1-3: for comparison purposes, it could be reasonable to show the measuring units for indicating the fraction either in vol% or in wt.%
- References: most of them are valuable papers coming from Chinese scientists: it could be reasonable to add some further references not related to Chinese works.
Author Response
Dear editor,
We are very grateful to your and the reviewers’ critical comments and thoughtful suggestions. Based on these comments and suggestions, we have made careful revision on the original manuscript entitled "Graphene-Based Thermal Interface Materials: An Application-Oriented Perspective on Architecture Design" (Manuscript ID: polymers-379385). We responded point by point to the comments of reviewer #2 as listed below, along with a clear indication of the location of the revision.
In specific response to the points raised by reviewer #2:
Question 1: Tables 1-3: for comparison purposes, it could be reasonable to show the measuring units for indicating the fraction either in vol% or in wt%.
Reply: Thank you for your comment. We have revised it. (See Table 1-3)
Question 2: References: most of them are valuable papers coming from Chinese scientists: it could be reasonable to add some further references not related to Chinese works.
Reply: Thank you for your suggestion. The references in this revision has been adjusted. (See Line 269-270, Page 8 and Line 309-311, page 9)
We appreciate for your warm work earnestly, and hope that the correction will meet with approval. The manuscript has been overall checked, and the changes marked in red. We hope that these revisions are sufficient to make our manuscript acceptable for publication in Polymers. If you believe that any additional clarifications need to be addressed, I will be happy to include them. Once again, thank you very much for your comments and suggestions.
Yours sincerely,
Cheng-Te Lin
Ningbo Institute of Materials Technology and Engineering, Chinese Academy of Sciences
E-mail: linzhengde@nimte.ac.cn

Reviewer 3 Report
Useful review, well-presented paper.
Abstract could be extended in order to clarify the specific employment the graphene-based TIMs are analyzed for.
Please correct "3b" at line 146.
Author Response
Dear editor,
We are very grateful to your and the reviewers’ critical comments and thoughtful suggestions. Based on these comments and suggestions, we have made careful revision on the original manuscript entitled "Graphene-Based Thermal Interface Materials: An Application-Oriented Perspective on Architecture Design" (Manuscript ID: polymers-379385). We responded point by point to the comments of reviewer #3 as listed below, along with a clear indication of the location of the revision.
In specific response to the points raised by reviewer #3:
Question 1: Abstract could be extended in order to clarify the specific employment the graphene-based TIMs are analyzed for.
Reply: Thank you for your comment. The abstract in this revision has been polished. (See Abstract)
Question 2: Please correct "3b" at line 146.
Reply: Sorry for the mistake. We have corrected it. (See Line 153, Page 8)
We appreciate for your warm work earnestly, and hope that the correction will meet with approval. The manuscript has been overall checked, and the changes marked in red. We hope that these revisions are sufficient to make our manuscript acceptable for publication in Polymers. If you believe that any additional clarifications need to be addressed, I will be happy to include them. Once again, thank you very much for your comments and suggestions.
Yours sincerely,
Cheng-Te Lin
Ningbo Institute of Materials Technology and Engineering, Chinese Academy of Sciences
E-mail: linzhengde@nimte.ac.cn
